# Role of the Anilinium Ion on the Selective Polymerization of Anilinium 2-Acrylamide-2-methyl-1-propanesulfonate

**DOI:** 10.3390/polym13142349

**Published:** 2021-07-17

**Authors:** Alain Salvador Conejo-Dávila, Marco Armando Moya-Quevedo, David Chávez-Flores, Alejandro Vega-Rios, Erasto Armando Zaragoza-Contreras

**Affiliations:** 1Department of Engineering and Materials Chemistry, Centro de Investigación en Materiales Avanzados, S.C., Miguel de Cervantes No. 120, Complejo Industrial Chihuahua, Chihuahua C.P. 31136, Mexico; alain.conejo@cimav.edu.mx (A.S.C.-D.); armando_10ncg@hotmail.com (M.A.M.-Q.); 2Facultad de Ciencias Químicas, Universidad Autónoma de Chihuahua, Chihuahua C.P. 31125, Mexico; dchavezf@uach.mx

**Keywords:** benzenaminium salts, free-radical polymerization, oxidative polymerization, polyaniline, selective polymerization, fluorescent polymer, π-stacking interactions

## Abstract

The development of anilinium 2-acrylamide-2-methyl-1-propanesulfonate (Ani-AMPS) monomer, confirmed by ^1^H NMR, ^13^C NMR, and FTIR, is systematically studied. Ani-AMPS contains two polymerizable functional groups, so it was submitted to selective polymerization either by free-radical or oxidative polymerization. Therefore, poly(anilinium 2-acrylamide-2-methyl-1-propanesulfonic) [Poly(Ani-AMPS)] and polyaniline doped with 2-acrylamide-2-methyl-1-propanesulfonic acid [PAni-AMPS] can be obtained. First, the acrylamide polymer, poly(Ani-AMPS), favored the π-stacking of the anilinium group produced by the inter- and intra-molecular interactions and was studied utilizing ^1^H NMR, ^13^C NMR, FTIR, and UV-Vis-NIR. Furthermore, poly(Ani-AMPS) fluorescence shows quenching in the presence of Fe^2+^ and Fe^3+^ in the emission spectrum at 347 nm. In contrast, the typical behavior of polyaniline is observed in the cyclic voltammetry analysis for PAni-AMPS. The optical properties also show a significant change at pH 4.4. The PAni-AMPS structure was corroborated through FTIR, while the thermal properties and morphology were analyzed utilizing TGA, DSC (except PAni-AMPS), and FESEM.

## 1. Introduction

Currently, there are two strategies for developing complex polymeric structures. The first approach is post-functionalization; however, it has the drawback that there is no way to control the degree of modification of the polymer [1,2]. The second synthesis pathway is monomeric design with a specific application [3,4]. In particular, monomers with two polymerizable sites or bifunctional monomers are the most used. If they contain polymerizable sites, they react by the exact polymerization mechanism, causing crosslinking [5,6]. However, when bifunctional monomers have different polymerizable groups, complex structures can be obtained. For example, Chan et al. [7] synthesized polythiophene phenylenes-g-polystyrene from a bifunctional monomer of modified thiophene phenylenes. Additionally, Klein et al. [8] reported developing a modified hydrogel through a bifunctional monomer that presents an epoxy functional group and maleimide substituents. Nevertheless, the synthesis route to develop these monomers is complicated and requires multiple steps. This study provides new insights into selective polymerization by designing a monomer of easy synthesis and polymerizable groups with different polymerization mechanisms. The use of these monomers will make it possible to build complex systems made up of two or more polymers with new properties or intrinsic characteristics of the system.

Additionally, our group recently reported the benzenaminium 4-styrenesulfonate salt. It was selectively polymerized to produce polyaniline or poly(benzenaminium 4-styrenesulfonate) [9]. This monomer has two aromatic rings that induce π-stacking interactions with itself [10]. The sulfonated aromatic fragment also provides hardness and stiffness to the polymer product obtained by the free-radical mechanism. It is important to note that the selective polymerization allowed the obtaining of core-shell nanoparticles, polystyrene-polyaniline, where the bifunctional monomer played the role of a bridge between both polymer phases [11].

The anilinium salts are produced by the acid-base reaction between aniline (pKa of 4.6) and a Brønsted-Lowry acid. Similarly, the Ani-AMPS monomer synthesis occurs under an identical reaction mechanism and involves versatile and easy-to-implement reactions; however, pH control is necessary according to the anion (conjugate base). The protonated nitrogen, also linked to the benzene ring, attracts the electronic density, allowing an alternate π-stacking conformation [12]. The benzenaminium cation affects the crystalline structure [13], solubility [14], and optical properties [15] of its salts due to the attraction forces (hydrogen bonding, van der Waals forces, coordination bonding) acting between them [10,15]. The physicochemical properties and the structural arrangement of the anilinium cation provide it with a variety of applications. For example, anilinium salts can be found in surfactants [16], ionic liquids [17], ionic semiconductors [18], lasers [15], organogels [19], and monomers, usually for polyaniline synthesis [20]. This variety of applications owes to the different types of the anion (organic or inorganic), typically analogous to sulfate and sulfonate, which have the function of cation anilinium stabilization.

Moreover, the oxidative polymerization of anilinium salts is a strategy to control doping in the PAni. The dopant determines the conductivity, electrochemical, and optical properties [11]. At a high concentration, the dopants do not significantly favor conductive or electroactive properties but increase the interaction of PAni with other compounds [21].

2-acrylamide-2-methyl-1-propanesulfonic acid (AMPS-acid) is an amphiphilic monomer analog of sulfonic acid, whose sodium salt is used as a flocculent and additive in paper production [22]. This compound is an aliphatic molecule that presents an amide functional group, allowing it to take different conformations and carry out hydrogen bonding depending on the solvent. The polymer derived from free-radical polymerization is poly(2-acrylamide-2-methyl propanesulfonic acid) (PAMPS-acid), a water-soluble and film-forming polymer, and is widely used as a polyelectrolyte to make hydrogels due to its high proton conductivity [23].

This paper reports the development of anilinium 2-acrylamide-2-methyl-1-propanesulfonate (Ani-AMPS). This new bifunctional monomer was submitted to selective polymerization (free-radical or oxidative polymerization), producing polymers with an altogether different structure. Particular emphasis was placed on how the anilinium group affects those properties. For example, compared with polyaniline doped with 2-acrylamide-2-methyl-1-propanesulfonic acid (PAni-AMPS), the poly(anilinium 2-acrylamide-2-methyl-1-propanesulfonic) [Poly(Ani-AMPS)] presented a selective turn-off fluorescent property in the presence of iron cations (Fe^2+^ and Fe^3+^) in an aqueous solution, whereas PAni-AMPS exhibited a significant change when studying the optical properties based on the pH and typical behavior of conducting polymers.

## 2. Materials and Methods

### 2.1. Materials

Aniline (Merck, Sigma-Aldrich, St. Louis, MO, USA >99.5%), prior to the experiment, was vacuum distilled, and ammonium persulfate (Merck, Sigma-Aldrich, St. Louis, MO, USA >98%), 2-acrylamide-2-methylpropane-1-sulfonic acid (Merck, Sigma-Aldrich, St. Louis, MO, USA >99%), sodium metabisulfite (Merck, Sigma-Aldrich, St. Louis, MO, USA >97%), acetone (Merck, Sigma-Aldrich, St. Louis, MO, USA), hydrochloric acid (Fermont, Monterrey, Nuevo León, México, 37%), ethanol (J.T. Baker, Phillipsburg, NJ, USA ), deuterium oxide (Merck, Sigma-Aldrich, St. Louis, MO, USA), methanol (J.T. Baker, Phillipsburg, NJ, USA), 1-propanol (J.T. Baker, Phillipsburg, NJ, USA), 2-propanol (J.T. Baker, Phillipsburg, NJ, USA), 1-butanol (J.T. Baker, Phillipsburg, NJ, USA ), and formic acid (Merck, Sigma-Aldrich, St. Louis, MO, USA) were used as received.

### 2.2. Synthesis of Anilinium Salt

Figure 1a illustrates the reaction for anilinium 2-acrylamide-2-methyl-1-propanesulfonate (Ani-AMPS) monomer synthesis. In a flat-bottomed flask (50 mL) equipped with a stirring bar and ice bath at 5 °C, 10 mL of an aqueous solution of AMPS-acid [1.93 M] was loaded. Next, aniline (19.3 mmol) was added dropwise under continuous magnetic stirring. Afterward, the flask was removed from the ice bath, and the reaction was left for 6 h at 25 °C. Subsequently, the water was evaporated at room temperature, and the solid was ground with a mortar. The solid was purified with two cycles of dispersion with acetone (25 mL) and filtration (yield 98%). ^1^H NMR (400 MHz, deuterium oxide) δ 7.64–7.54 (m, 3H), 7.45 (dt, *J* = 8.0, 1.5 Hz, 2H), 6.26 (dd, *J* = 17.1, 10.1 Hz, 1H), 6.16 (dd, *J* = 17.1, 1.5 Hz, 1H), 5.72 (dd, *J* = 10.1, 1.5 Hz, 1H), 3.44 (s, 2H), 1.53 (s, 6H); ^13^C NMR (101 MHz, Deuterium oxide) δ 197.79, 130.99, 130.11, 129.75, 129.20, 126.57, 122.85, 57.15, 51.98, 26.36 ppm, see Appendix A; FTIR (ATR) v_max_/cm^−1^ 3276, 2875, 2638, 1654, 1620, 1177, 1157, 1041.

### 2.3. Free-Radical Polymerization of Ani-AMPS

The free-radical polymerization of Ani-AMPS, shown in Figure 1b, was performed in a flask equipped with a stirring bar, temperature control, and nitrogen atmosphere. Aqueous solutions of Ani-AMPS [75 × 10^−3^ M] and sodium metabisulfite [3.34 × 10^−3^ M] were added together (volume total 24 mL). At this point, the pH value was near 4. The monomer solution and initiator were nitrogen bubbled for 20 min and heated to 60 °C. At this time, 6 mL of an aqueous solution of ammonium persulfate [6.67 × 10^−3^ M] was injected into the reactor to launch the polymerization. The polymerization was left for 6 h under a nitrogen atmosphere and continuous stirring. When the polymerization finished, the water was eliminated at room temperature. The poly(anilinium 2-acrylamide-2-methyl-1-propanesulfonate), poly(Ani-AMPS), was purified under the precipitation technique (three times), dissolved in water (5 mL) and acetone (25 mL) as non-solvent. The product was filtered, and residual acetone evaporated at room temperature (yield 95%). Spectral data: ^1^H NMR (400 MHz, deuterium oxide) δ 7.80–7.29 (m, 10H), 3.71 (t, *J* = 6.3 Hz, 1H), 3.49–3.25 (m, 4H), 2.67 (t, *J* = 6.3Hz, 1H), 2.10 (m, 2H), 1.82–1.55 (m, 2H), 1.53–1.43 (m, 12H); ^13^C NMR (101 MHz, Deuterium Oxide) δ 170.89, 167.79, 133.93, 130.98, 130.43, 130.14, 129.65, 129.67, 126.56, 122.94, 122.69, 57.12, 56.59, 51.97, 51.87, 47.73, 31.30, 26.55, 26.37, 26.35 ppm; see Appendix A; FTIR (ATR) v_max_/cm^−1^ 3300, 3072, 2971, 2614, 1657, 1552, 1177.

### 2.4. Oxidative Polymerization

Figure 1c shows the oxidative polymerization of Ani-AMPS. Firstly, an aqueous solution of the monomer [12 × 10^−2^ M] was prepared. In another graduated vessel, the ammonium persulfate [15 × 10^−2^ M] solution was formulated. Both solutions (total volume 50 mL) were poured into a 250 mL vessel and mixed thoroughly for 5 min. The polymerization was left for 24 h, at 4 °C. Afterward, the byproducts of the polymerization were removed with cycles (three times) of centrifugation (5000 rpm for 10 min) and washing with an HCl-ethanol solution (5 vol%). The polymer product was recovered by evaporation at room temperature (yield 95%). FTIR (ATR) v_max_/cm^−1^ 3204, 2838, 1655, 1584, 1563, 1488, 1296, 810.

### 2.5. Characterization

The characterization by proton nuclear magnetic resonance (^1^H NMR) and carbon nuclear magnetic resonance (^13^C NMR) of Ani-AMPS and Poly(Ani-AMPS) was performed using a Brucker spectrometer (NMR Bruker Ascend 400 MHz, Billerica, MA, USA) at 400 MHz, 7.05 T, and 25 °C. The samples were analyzed utilizing deuterium oxide as the solvent reference. Spectra were acquired in 1.73 s, employing 16 scans. The structural characterization of the products was also confirmed by infrared spectroscopy (GX-FTIR, PerkinElmer, Waltham, MA, USA) using the ATR accessory. Spectra were studied in the region from 4000 to 400 cm^−1^, performing 30 scans, with a resolution of 4 cm^−1^. The molecular weight of Poly(Ani-AMPS) was determined by employing a gel permeation chromatographer (1260 Infinity, Agilent, Santa Clara, California, USA) with a refractive index detector. The equipment had three columns in molecular weight ranges from 100 to 30,000 g/mol, 10,000 to 200,000 g/mol, and 50,000 to 1,000,000 g/mol. The mobile phase was a pH 8 buffer solution (NaNO_3_ 0.2 M, NaH_2_PO_4_ 0.01 M, NaN_3_ 100 ppm). Ten standards samples of poly(ethylene glycol) (M_p_ = 106 g/mol lower limit, M_p_ = 21,300 g/mol higher limit) were used for calibration purposes. Fluorescence properties from Poly(Ani-AMPS) were performed on a fluorescence spectrometer (PerkinElmer LS-45, Waltham, MA, USA). In addition, the concentrations of Poly(Ani-AMPS) and binary metallic salts were 500 ppm and 1 mM, respectively. The study of optical properties was achieved by UV-Vis-NIR (Evolution 220, Thermo Fisher Scientific, Waltham, MA, USA) spectroscopy. The thermal degradation of the samples (10 mg) was characterized using a thermal analyzer (SDT Q600, TA Instruments, New Castle, DE, USA) under air atmosphere. Measurements were performed at a heating rate of 10 °C min^−1^ from room temperature to 700 °C. A differential scanning calorimeter (DSC Q2000, TA Instruments, New Castle, DE, USA) was utilized to measure the glass transition temperature (T_g_) in order to contrast the Poly(Ani-AMPS) value with Poly(AMPS-acid). Specifically, 10 mg of sample was used under a nitrogen atmosphere at a heating rate of 10 °C min^−1^. The morphological analyses of Poly(Ani-AMPS) and PAni-AMPS were performed using a field emission scanning electron microscope (FE-SEM, JSM-7401F, JEOL, Akishima, Japan) equipped with an r-filter that allowed variable energy to filter secondary electrons and backscattered electrons.

The electrochemical characteristics of PAni-AMPS were determined utilizing a potentiostat/galvanostate analyzer (Model 1260 plus 1287, Solartron Analytical, Farnborough, Hampshire, UK). The study was conducted with a standard three-electrode cell, where a platinum plate (1 cm^2^), an Ag/AgCl/saturated KCl electrode, and carbon paste plus a sample film were the counter electrode, reference electrode, and working electrode, respectively. The electrolyte utilized was a sulfuric acid solution [2.0 M]. The potential window was from −0.5 to +1.0 V versus Ag/AgCl/saturated KCl electrode at a scan rate of 10 mV s^−1^.

## 3. Results and Discussion

The selective polymerization that occurs in Ani-AMPS monomer depends mainly on the polymerization conditions. For instance, in free-radical polymerization, a redox initiator is utilized, where sodium metabisulfite (MBS) is first added to produce sodium bisulfite in the presence of water. Thus, salt reacts with ammonium persulfate (APS) to generate free radicals, initiating the polymerization process. It is important to note that sodium bisulfite has a lower oxidation potential than anilinium salts, so APS reacts with sodium bisulfite. In contrast, APS acts as an oxidative agent in oxidative polymerization because the system does not contain MBS. In addition, oxidative polymerization, specifically for PAni-AMPS, requires a molar ratio of 1.25:1 (ammonium persulfate:Ani-AMPS). Compared to free-radical polymerization, oxidative polymerization demands an increased amount of initiator as an oxidizing agent. Appendix A illustrates the scheme of the decomposition of MBS and its reaction with APS.

### 3.1. Monomer Characterization

The Ani-AMPS monomer synthesis occurs through an acid-base reaction between AMPS-acid and aniline. For structural elucidation, ^1^H NMR and FTIR spectroscopy were employed; Figure 2A shows the ^1^H NMR spectrum of Ani-AMPS. The signals at 7.64–7.45 ppm correspond to phenyl protons (type AB_2_X_2_ system), which are characteristic of monosubstituted benzene [24]. Specifically, the H_b_, H_c_ (m, 7.64–7.54 ppm, 3H), and H_d_ (dd, 7.45 ppm, 2H) protons. The signals of the vinyl protons (system AMX) are H_e_ (dd, 6.26 ppm, 1H, J_e-f and e-j_ = 17.1, 1.5 Hz), H_f_ (dd, 6.16, 1H, J_f-j, f-e_ = 17.1, 10.1 Hz), and H_j_ (dd, 5.72 ppm, 1H, J_j-f, j-e_ = 10.1, 1.5 Hz) [25]. The spectra also present two more peaks, including the H_g_ proton (s, 3.44, 2H) attributed to the α-methylene sulfonate group and, finally, the H_i_ proton (s, 1.53, 6H) was assigned to the γ-methyl sulfonate group [26].

Figure 2B shows the FTIR spectra of AMPS-acid, aniline, and Ani-AMPS. Compared with AMPS-acid and aniline, the Ani-AMPS presents characteristic bands that validated the anilinium cation formation, specifically at 2638 cm^−1^ and 1157 cm^−1^, associated, respectively, with an overtone and stretching vibration of the ^+^N-H and C-N^+^ bond [15]. It is essential to note the significant change in the vibration corresponding to C-N of the Ani-AMPS to its C-N^+^ form present in the polymer. Additionally, the peaks of AMPS anion functional groups are present in the monomer, such as the secondary amide exhibiting a band at 3276 cm^−1^ due to N-H stretching vibrations [27]. Notably, in this region (3370–3270 cm^−1^), the amides in a solid or liquid state may have bands due to hydrogen bonds. 

For the methyl groups, the symmetric stretching and asymmetric deformation vibrations of C-H occur near 2875 cm^−1^ and at 1465 cm^−1^, respectively. The stretching vibration corresponding to the carbonyl group is shown at 1654 cm^−1^. The alkene absorption (type stretching vibration) band occurs at a wavenumber of 1620 cm^−1^. Additionally, the spectrum presented bands at 1041 cm^−1^ and 1177 cm^−1^, correlated, respectively, with the symmetric stretching vibration of S-O and symmetric flexion of S=O of the SO_3_ group [28]. Based on this characterization, the formation of Ani-AMPS can be confirmed.

### 3.2. Characterization of the Product of Free-Radical Polymerization

The free-radical polymerization of the Ani-AMPS occurs over the vinyl group, forming an aliphatic chain. The polymer product, Poly(Ani-AMPS), was characterized by ^1^H NMR and ^13^C NMR spectroscopy to confirm the polymer structure, as shown in Appendix A. Figure 3a illustrates the ^1^H NMR of Poly(Ani-AMPS) utilizing deuterium oxide as a solvent. It is interesting to note that the ^1^H NMR spectrum integration and the number of signals corresponding to ^13^C NMR are equivalent to two monomers, as shown in Appendix A. Furthermore, the spectrum also presents four new signals at a high field associated with H_e_, H_f_, and H_j_ protons, due to the hybridization change from sp^2^ to sp^3^ caused by the polymerization process [26]. The proton (H_e_) has two identical triplets at 3.59 ppm (1H) and 2.55 ppm (1H) assigned like H_e_ and H_e*_ (only to differentiate protons), respectively. The H_f_ and H_j_ protons are presented as broad bands located at 1.98 ppm (2H) and 1.62 ppm (2H), corresponding to H_f_ + H_j_ and H_f*_ + H_j*_, respectively. Therefore, the similarity in the shape, pattern, and integration of these signals suggests the formation of a complex structure. The spectrum also shows a multiplet and doublet (7.55–7.25 ppm, 10H) corresponding to aromatic protons of two anilinium cations. Compared with the Ani-AMPS, Poly(Ani-AMPS) aromatic proton signals appear at a high field close to 0.2 ppm. These changes, specifically in the region of the aromatic ring peaks, are due to π-stacking interactions, forming more complex structures known as self-assembly. Several reports on this effect employed different strategies, especially the increase in the number of aromatic ring substitutes and the concentration concerning π-stacking. Consider, as an example, the self-association between pyridine and benzene functionalized with bis-pyrene methyl amide, induced by hydrogen bonding and π interactions, according to Kim et al. [29]. Another path, π-stacking, was evident by the shift from aromatic protons to higher fields when the sample concentration was studied based on these interactions [30].

The structural validation of Poly(Ani-AMPS) was corroborated by FTIR spectroscopy, compared with Ani-AMPS. Figure 3b shows the Ani-AMPS and Poly(Ani-AMPS) spectra. The Poly(Ani-AMPS) has substantially decreased molecular vibrations, particularly in the aromatic fingerprint, suggesting π-stacking interactions. For instance, the aromatic C-H out-of-plane deformation vibrations at 865, 825, and 690 cm^−1^. The C-C stretching vibrations of the aromatic ring also have an identical effect, particularly at 1575 and 1475 cm^−1^. Additionally, Ovchinnikov et al. reported the formation of π-stacking interaction and self-association of methylene blue in the region at high vibrational frequency [31]. Specifically, a significant change was observed regarding the shape of the absorption band at 3072 cm^−1^ for Poly(Ani-AMPS), ascribed to C-H stretching vibrations of the aromatic group. Lastly, to complement Poly(Ani-AMPS) characterization, the molecular weight was determined by GPC. Mw of 9345 g/mol and a polydispersity index of 4.39 were obtained. It is important to note that this molecular weight corresponds to the polyanion due to the pH of the mobile phase.

Moreover, the C-C stretching (1620 cm^−1^) and C-H deformation vibrations concerning the alkene functional group decreased because of polymerization. Particularly, near 913 cm^−1^, corresponding to CH_2_ out-of-plane deformation vibration, and the region from 970–940 cm^−1^, corresponding to C-H out-of-plane deformation vibration [32]. Additionally, the C-H (methyl group) asymmetric stretching vibration occurs at 2976 cm^−1^. For the α-methylene sulfonate group, two bands are observed at 2926 cm^−1^ and about 2870 cm^−1^, corresponding to the asymmetric and symmetric C-H stretching vibration, respectively. Further, an overlap (around 2870 cm^−1^) corresponding to the symmetric stretching vibration band for the methyl and α-methylene sulfonate group is observed. A shift of about 21 cm^−1^ at most is observed for the ^+^N-H bond (anilinium group) in the Poly(Ani-AMPS) spectrum. 

A broad band is observed around 1650 cm^−1^ because the Poly(Ani-AMPS) sample contained water that was difficult to remove, so the peaks corresponding to the stretching and flexion vibration for C=O (1654 cm^−1^) and N-H (1552 cm^−1^) are not defined as in the Ani-AMPS monomer [33].

### 3.3. Photoluminescence Properties of Poly(Ani-AMPS)

The fluorescence of the materials could be modified in the presence of metals, anions, or cations, or the change in solvent polarity [34]. For example, Figure 4a shows the spectrum of Poly(Ani-AMPS) in an aqueous solution at 550 ppm. The Poly(Ani-AMPS) presents two excitation peaks with a maximum at 278 nm and the emission peak at 347 nm, corresponding to π-π* transitions of the aromatic ring (anilinium cation) [35].

The fluorescent properties of Poly(Ani-AMPS) were investigated when analyzing the interaction between the aromatic ring (anilinium cations) and chloride salt. It is worth saying that using a fluorescent sensor is a convenient method to analyze metal ions due to their high sensitivity and easy operability [36,37]. To analyze the effect of different metal ions on the fluorescence of the polymer, several aqueous solutions were prepared utilizing chloride salts (Al^3+^, Na^+^, Mg^2+^, Li^+^, K^2+^, Cu^2+^, Ca^2+^, Fe^2+^, Fe^3+^, NH^4+^) as solutes at 1 mM concentration. The study was based on the reduction of the emission peak, evidencing the interaction with binary salt. Figure 4b displays the percentage of quenching with the different ions analyzed. The Poly(Ani-AMPS) presented a selectivity quenching for Fe^2+^ and Fe^3+^ [36]. The fluorescence turn-off was attributed to coordination interactions between aromatic rings (anilinium cations) and Fe^2+^ or Fe^3+^ [38]. To summarize, from a quantum mechanics point of view, it suggests that it takes precedence over orbital *d* cations under these conditions. It is important to note that iron cations play a fundamental role in biological and environmental processes.

The detection limit for Fe^2+^ and Fe^3+^ was determined at several concentrations. Figure 5a,c illustrate the abatement in the fluorescence signal of the emission spectrum and the correlation of maximum intensity at 348 nm versus concentration for Fe^2+^ and Fe^3+^. The limit of detection is 500 μM and 20 μM, respectively, for Fe^2+^ and Fe^3+^. The behavior, in general, presented two linear zones with different concentration intervals. Figure 5b shows the fluorescence intensity of the emission spectrum regarding the concentration of Fe^2+^. The first and second zone are located from 0 μM to 1000 μM and from 1000 μM to 4000 μM. Nevertheless, for the Fe^3+^ system, the Poly(Ani-AMPS) has significant selectivity compared with the analogous Fe^2+^. Figure 5b displays the two linear zones, where the first is from 20 μM to 80 μM, and the second region is from 100 μM to 400 μM [39]. Appendix A displays the values of fluorescence intensity concerning concentration for Fe^2+^ and Fe^3+^. The different behavior of both cations is due to solubility in water (Fe^3+^ >> Fe^2+^), with the solvent playing an important role when sensing [40].

The sensitivity of the polymer is comparable with previous reports; however, its detection range is higher than most of the reported ranges [37,41,42]. For example, Chen et al. reported that the synthesis of polyphosphazene microspheres has selectivity for Fe^3+^ aqueous solutions and a detection limit of 76 × 10^−9^ M (0.076 μM) in a range from 0 to 10 μM [37]. Additionally, Yang et al. reports sensitivity for Fe^3+^ in *N*, *N* dimethylformamide solutions, presenting a detection limit of 0.21 μM [0.21 to 150 μM], using Eu-based coordination polymer, including 4-(pyridyl-N-oxide) methylphosphonic acid as a ligand [43].

### 3.4. Oxidative Polymerization

The Ani-AMPS monomer can be polymerized through an oxidative mechanism, producing polyaniline doped with 2-acrylamide-2-methyl-1-propanesulfonate (PAni-AMPS). Contrary to the Poly(Ani-AMPS) structure, PAni is the polymer backbone, and AMPS anion is the dopant. The analysis of chemical structures and electroactive properties of PAni-AMPS was confirmed by FTIR spectroscopy and cyclic voltammetry, as shown in Figure 6. Typical molecular vibrations reported for PAni are observed; for example, 1584 cm^−1^ and 1488 cm^−1^ were assigned to the stretching vibrations of the C=N bond of the quinoid ring and the C-N bond of the benzenoid ring, respectively [44,45]. The peak height of the benzoid group is higher than the quinoid structure, suggesting the presence of PAni in the form of emeraldine salt [46]. The spectrum also presents a signal at 810 cm^−1^ assigned to the *p*-benzene ring disubstituted position. These three signals confirmed the backbone formation [47]. In addition, the spectrum shows the characteristic peaks of AMPS anion, for instance, at 1664 cm^−1^ and 1552 cm^−1^ described previously for the amide group. The presence of these signals indicates the formation of PAni doped with the AMPS anion [48].

The electroactivity is characteristic in conductive polymers; consequently, PAni-AMPS was studied. Figure 6b shows the cyclic voltammogram of PAni-AMPS utilizing sulfuric acid [1 mM] as the electrolyte, in the potential window from −0.5 to 1.0 V. As noted, three oxidation signals at 0.325 (A), 0.62 (B), and 0.86 V (C) are present. The peaks A/A1 and C/C1 were attributed to the redox transitions leucoemeraldine/emeraldine and emeraldine/pernigraniline, respectively [49]. Specifically, peak A presents an increase and displacement towards positive potential, comparing the first against the fifth cycle. Additionally, this minor change is related to the exchange of dopants between the AMPS and the electrolyte dopant agent (HSO_4_)^−^ [50]. The transition emeraldine/pernigraniline (peak C) decreased relative to the number of cycles due to the doping exchange, overlapping with peak B, causing an increase in current density. Peak B, in the fifth cycle, shows a maximum current density at 0.68 V. Compared to peak B, the oxidation peak of PAni-(HSO_4_)^−^ for transition emeraldine/pernigraniline was reported at 0.72 V, confirming dopant exchange [51]. Additionally, dimers or polymer hydrolysis formation owing to the electrolyte is observed at peak B/B1 [52].

Several applications of electroactive PAni, such as in electrodes [53], supercapacitors [54], sensors [55], conductive inks, anticorrosive coatings, and conductive surfaces [56,57], have been demonstrated. Therefore, PAni-AMPS has the potential for application in these areas.

### 3.5. Comparison of Poly(Ani-AMPS) and PAni-AMPS

#### 3.5.1. Optical Properties

The role of the anilinium cation in the structure of the two polymers is altogether different, e.g., for Poly(Ani-AMPS), it acts as the counterion, and for PAni-AMPS, it constitutes the backbone. These significant changes in basic chemical structure reflect differences in optical properties, according to UV-Vis-NIR spectroscopy. Figure 7a shows the UV-Vis-NIR spectra of Ani-AMPS, Poly(Ani-AMPS), and PAni-AMPS in an aqueous solution at a concentration of 660 ppm. The spectrum of Ani-AMPS presents a shoulder signal at 285 nm, corresponding to π-π* of the aromatic ring (anilinium cation). Compared with Ani-AMPS, the Poly(Ani-AMPs) shows two signals at 285 and 352 nm. Similarly to Ani-AMPS, the shoulder signal at 285 nm was assigned to π-π* of the aromatic ring [58]. Furthermore, the band at 352 nm was attributed to the π-stacking effect that occurs in anilinium cations through the aromatic ring, forming the Poly(Ani-AMPS) complex structures [10,14,15]. It should be noted that the shoulder observed for the monomer and its polymer close to 285 nm was confirmed with the first derivative of each spectrum, as shown in Figure 7a.

Moreover, the PAni-AMPS spectrum, shown in Figure 7a, presents the typical bands of emeraldine salt, i.e., at 363 nm assigned to the benzenoid structure, at 442 nm to the polaron transition, and finally, a broad band from 700 to 1000 nm, where the bipolar transition at 790 nm is observed [59,60]. The color of the PAni-AMPS is dark green, in the state of emeraldine salt, produced by polaron-bipolaron equilibrium, and also gives the conductive properties to polyaniline [61].

Polyaniline is employed as a colorimetric pH sensor based on a fundamental chemical principle causing the removal of the dopant agent and, therefore, a polymer color change. In addition, the point where color alteration occurs is associated with pKa of doped polyaniline, which indicates the loss of the dopant agent. To determine the pH at which the PAni-AMPS doping-dedoping (color change) occurs, UV-Vis-NIR spectra were obtained from pH 1 to 12, as shown in Figure 7b. It is clear that the polaron and bipolaron signals are present at an acid pH of less than 4. At pH 5, the polaron signal intensity decreases significantly, and the broad band of bipolaron starts to shift to blue. When the polaron signal presents half of the intensity concerning pH 1 (maximum) and pH 12 (minimum), the pKa for the PAni-AMPS was determined at 4.4, as shown in Figure 7b. The spectra from pH 6 to 12 show that the broad band is located from 500 to 900 nm with a shoulder peak at ~660 nm assigned to the quinoid ring of the emeraldine base, the nonconductive form of polyaniline, with the characteristic blue color.

The π-stacking effect has been investigated in several molecules, analyzing solvent type, concentration, and time, among other variables [62,63,64]. Additionally, UV-Vis spectroscopy is one of the characterization techniques for identifying the presence of π-stacking [65]. This effect in Poly(Ani-AMPS) is produced by the inter- or intra-chain or both interactions among the anilinium cations, AMPS anion, and solvent. The equilibrium forces between the AMPS anion and anilinium cation (van der Waals forces, electrostatic repulsion, steric hindrance) allow the interaction and the final conformation, too [62,66].

Previously, a polar solvent was selected with different functional groups, such as alcohol, ketone, and carboxylic acid. The solvents selected were acetone, ethanol, formic acid, and water, which present a dielectric constant of 20.7, 36.4, 58, and 80.4, respectively. The purpose was to find whether the solvent types keep the cation anilinium stable with the polymer. Figure 8a illustrates the UV-Vis-NIR spectra of solutions of Poly(Ani-AMPS) in these solvents at a concentration of 660 ppm. For acetone, the spectrum shows bands at 264, 286, and 306 nm, assigned to the transitions n-π* of the nitrogen of the anilinium cation, π-π* of the aromatic ring, and n-π* of oxygen in the acetone, respectively. The last signal was interpreted as the electrostatic interaction between the polymer and acetone. The spectra of water and ethanol were similar, only presenting a band around 285 nm, attributed to the aromatic ring of the anilinium cation. On the other hand, formic acid demonstrates a band at 254 nm, associated with the n-π* transition of oxygen from the carboxylic group; consequently, the absence of the aromatic ring suggested that the acid media protonated the polymer, producing counterion exchange, contrary to water or ethanol, that did not interact or modify the polymer. In summary, Poly(Ani-AMPS) stability, specifically the anilinium cation, is not affected by alcohol or water.

The effects of alcohol-type solvents and water on the π-stacking of the Poly(Ani-AMPS) were studied. Figure 8b shows the spectrum of Poly(Ani-AMPS) solution in water and several alcohols, i.e., methanol, ethanol, 1-propanol, 2-propanol, and 1-butanol, at a concentration of 660 ppm. In general, UV-Vis-NIR spectra of alcohol solutions presented a hyperchromic effect at 352 nm compared with the analog of the aqueous solution. This behavior is due to increased interactions of the anilinium cation caused by the dielectric constant of the alcohols, producing the π-stacking effect. The butanol solution presented the highest absorbance at 352 nm and therefore stimulated the π-stacking effect. Hence, alcohols with a lower dielectric constant present stronger π-stacking interactions than alcohols with higher values [67].

The physical interactions that favored the π-stacking conformation are related to the concentration of the solution and time. Figure 8c illustrates the UV-Vis spectra of Poly(Ani-AMPS) solution in butanol at different concentrations, from 130 to 6600 ppm, for 4 h. It is evident that the band at 352 nm depends solely on the polymer concentration, and hence the formation of π-stacking interactions. Additionally, the dimer generation is attributed to the band that appears at 560 nm, particularly above 1300 ppm. Petr et al. reported that at a concentration above 1300 ppm, a band at 555 nm is observed, attributed to the oxidation process of anilinium cations to form dimers or oligomer analog N1-phenylbenzene-1,4-diamine [68].

Moreover, Figure 8d shows the spectra at two analysis times (4 and 144 h) when employing butanol and water as the solvents, at a concentration of 660 ppm. Compared with Poly(Ani-AMPS) solution in water (4 h and 144 h), the Poly(Ani-AMPS) solution in butanol, under identical conditions, has absorbance that is 2.2- and 4-fold higher, for 4 h and 144 h, respectively. In the case of the water system, there are insignificant changes when comparing 4 h against 144 h. On the contrary, the Poly(Ani-AMPS) solution in butanol increased about 2-fold. Therefore, the concentration and time increment, especially for butanol, stimulated chain interactions and the π-stacking effect on Poly(Ani-AMPS).

#### 3.5.2. Thermal Analysis

Figure 9 shows the thermogravimetric traces of Ani-AMPS, Poly(Ani-AMPS), and PAni-AMPS. The thermal degradation of Ani-AMPS, shown in Figure 9a, occurred in three steps. The first weight loss between 170 and 250 °C, corresponding to 31%, was assigned to anilinium ion degradation [15,69]. The second drop, between 250 and 340 °C, corresponded to desulfonation [70], and the last transition was assigned to vinyl-amide thermal degradation, occurring between 340 and 540 °C [71].

Figure 9b shows the PAni-AMPS thermogram, which exhibited two main characteristics of weight transitions, like classical doped polyaniline [72]. First, the two thermal degradation steps from 160–360 °C and 360–625 °C were assigned, respectively, to the AMPS anion loss, corresponding to 46%, and the backbone decomposition, corresponding to 43% [73]. By adjusting these experimental values to 100%, PAni and AMPS anion have a percentage ratio of 48% and 52%, respectively. Second, in a precise definition, based on the chemical structure of emeraldine salt, the basic unit consists of four aromatic rings at the main chain and two anions, acting as dopants. Thus, the calculated molecular weight of the basic unit of PAni-AMPS is 777.96 g mol^−1^. In addition, the AMPS anion has a molecular weight of 206.24 g mol^−1^, meaning that the dopant accounts for 53% of the basic unit of PAni-AMPS. Therefore, the percentages obtained experimentally are similar to those calculated, which confirmed the structure of PAni doped with AMPS anion.

Moreover, Ani-AMPS and Poly(Ani-AMPS) showed two transitions close to 210 °C and 300 °C, assigned to the anilinium cation loss and desulfonation. A third transition close to 425 °C was also present in Ani-AMPS and Poly(Ani-AMPS), which corresponded to a weight loss of 20%, which was assigned to the decomposition of the amide substituent to form a nitrogen derivative. Finally, Poly(Ani-AMPS), shown in Figure 9c, presented the last transition at 530 °C, ascribed to the thermal decomposition of the polymer backbone [22], corresponding to 19%. As observed, the thermal decompositions of Ani-AMPS and Poly(Ani-AMPS) have differences in transitions occurring at temperatures above 400 °C. 

To supplement the thermal analysis, Poly(Ani-AMPS) was characterized by differential scanning calorimetry (DSC). Figure 9d shows the thermograms of Poly(Ani-AMPS) and PAMPS-acid (the acid form of Poly(Ani-AMPS)). PAMPS-acid presented a T_g_ of 109 °C according to the literature [22]. Similarly, Poly(Ani-AMPS) presented a T_g_ of 110 °C, suggesting that in the solid state, π-stacking interactions between anilinium cations of Poly(Ani-AMPS) did not affect the polymer chain flexibility.

#### 3.5.3. Morphology

Figure 10a,e shows the morphology of Poly(Ani-AMPS) and PAni-AMPS. As observed, the polymers presented entirely different morphologies. Poly(Ani-AMPS) formed a film with some dark domains, ascribed to an increment in the electron density, owing to anilinium ion π-stacking interactions [74]. These domains showed either vesicle-like (spheres) or lamellar-like (lines) morphology. On the other hand, PAni-AMPS showed fibrillary morphology [75], commonly observed in diluted aniline polymerizations [76]. Likewise, the morphology of polyanilines depends on polymerization methodology [77].

Figure 10b,c show the physical aspect of Poly(Ani-AMPS) in solid-state and aqueous solutions. Furthermore, Poly(Ani-AMPS) was entirely water soluble and formed a yellow translucent film over the substrate. Meanwhile, PAni-AMPS, shown in Figure 10d,f, is a green powder in the solid state, typical of emeraldine salt. The aqueous phase forms an unstable suspension that precipitates after some minutes. The observed differences in both morphology and behavior are a consequence of the characteristic molecular structure of each polymer.

## 4. Conclusions

The design and development of anilinium 2-acrylamide-2-methyl-1-propanesulfonate (Ani-AMPS) monomer allows the synthesis of complex polymeric systems with different attributes to conventional polymers. The Ani-AMPS monomer was submitted to selective polymerization by free-radical polymerization and oxidative polymerization, producing poly(anilinium 2-acrylamide-2-methyl-1-propanesulfonic) [Poly(Ani-AMPS)] and polyaniline doped with 2-acrylamide-2-methyl-1-propanesulfonic acid PAni-AMPS, respectively. The position of the anilinium cation as a counterion (Poly(Ani-AMPS)) or as part of the polymer backbone (in PAni-AMPS) played an essential role in the optical properties of each polymer. For Poly(Ani-AMPS), π-π* transition of aromatic rings favored a π-stacking conformation by the presence of the 2-acrylamide-2-methylpropane-1-sulfonate that links the polymer main chain with the anilinium cation. The π-stacking conformation gives the polymer different absorption bands and intensities, depending on the solvent and polymer concentration. Notably, the photoluminescence properties of Poly(Ani-AMPS), water solubility, and the simplicity of the methodology could be employed to develop a fluorescent sensor to identify water-soluble iron ions. Meanwhile, for PAni-AMPS, although having the same π-π* transitions, the conjugation of the aromatic rings is increased by the polymerization process. The absorption bands in this polymer are related to the presence or absence of the dopant. In this case, dedoping occurred above pH 4.4. These structural differences were also observed in thermal stability since each polymer presents different thermogravimetric traces. As for the morphology, Poly(Ani-AMPS) produced a translucent film, whereas the PAni-AMPS formed water-insoluble short nanofibers. Additionally, in Poly(Ani-AMPS), the anilinium ion opens the possibility to graft a conducting polymer, by oxidative polymerization, over its surface to design conductive inks.

## 5. Patents

Mx/a/2019/015137 “Process of synthesis of polyfunctional comonomers and their polymerization”.

## Figures and Tables

**Figure 1 polymers-13-02349-f001:**
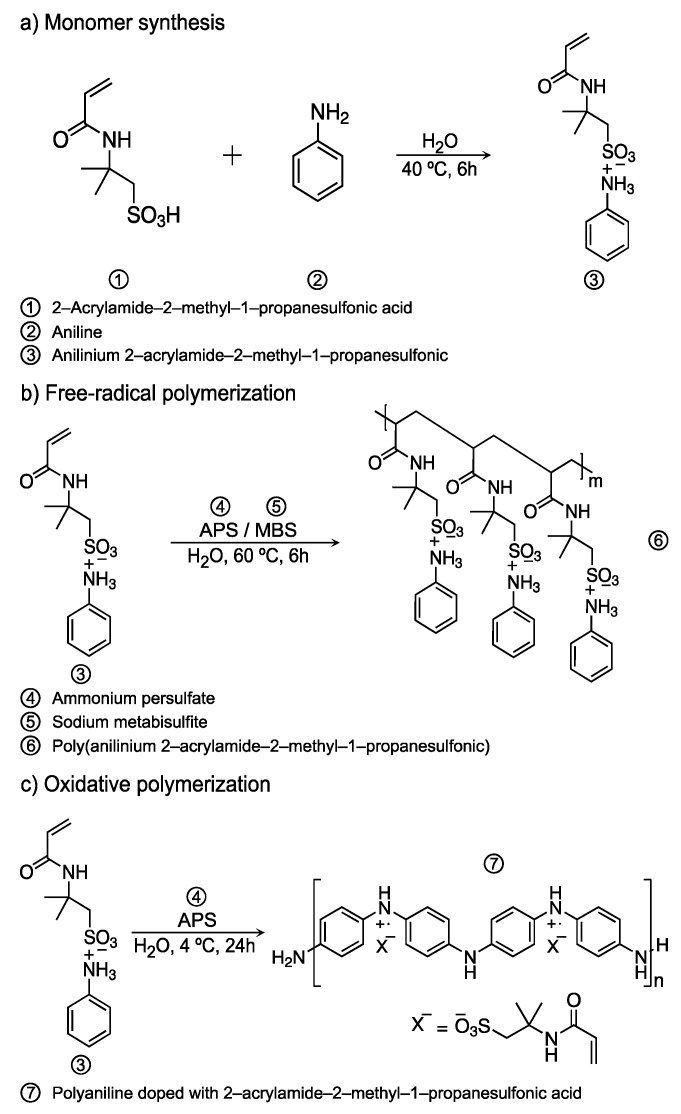
The synthesis scheme of (**a**) Ani-AMPS; (**b**) free-radical polymerization of Ani-AMPS monomer; and (**c**) oxidative polymerization of Ani-AMPS monomer.

**Figure 2 polymers-13-02349-f002:**
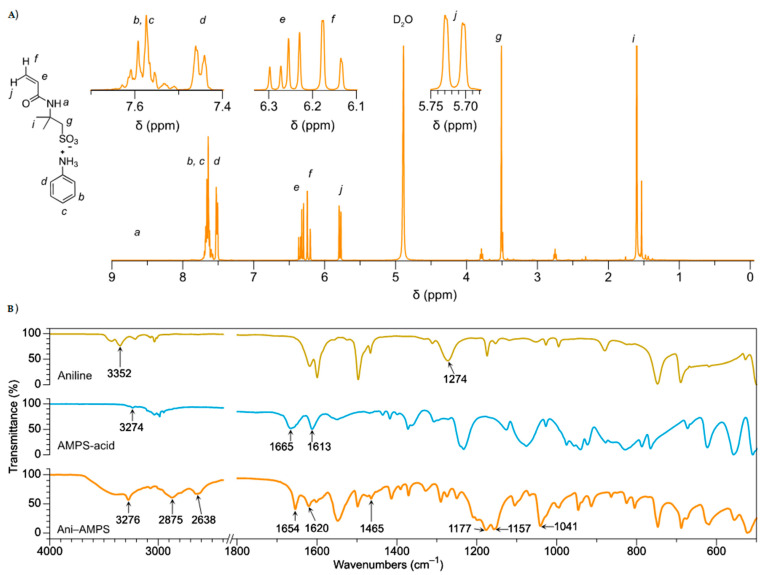
Ani-AMPS monomer spectra: (**A**) ^1^H NMR (solvent: deuterium oxide) δ 7.64–7.54 (m, 3H), 7.45 (dt, *J* = 8.0, 1.5 Hz, 2H), 6.26 (dd, *J* = 17.1, 10.1 Hz, 1H), 6.16 (dd, *J* = 17.1, 1.5 Hz, 1H), 5.72 (dd, *J* = 10.1, 1.5 Hz, 1H), 3.44 (s, 2H), 1.53 (s, 6H) and (**B**) FTIR spectra, comparison of Ani-AMPS versus aniline and AMPS-acid.

**Figure 3 polymers-13-02349-f003:**
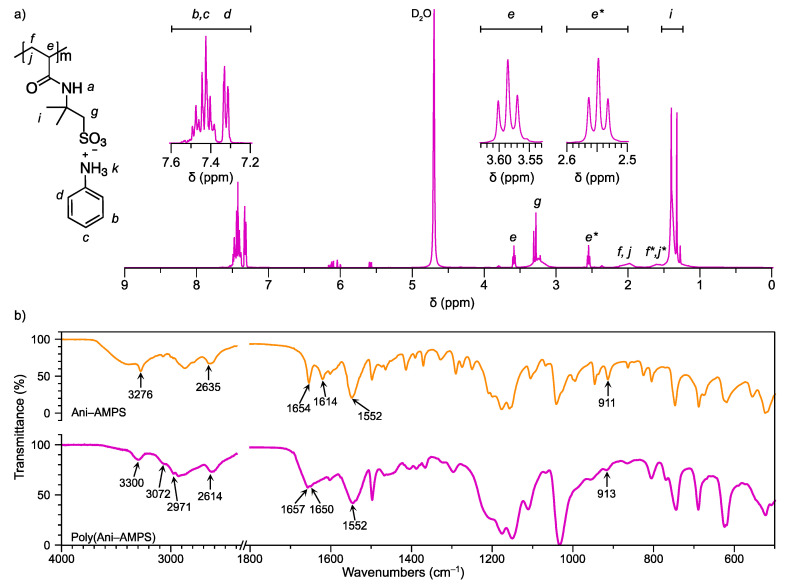
Poly(Ani-AMPS) spectra: (**a**) ^1^H NMR (solvent: deuterium oxide) δ 7.80–7.29 (m, 10H), 3.71 (t, *J* = 6.3 Hz, 1H), 3.49–3.25 (m, 4H), 2.67 (t, J = 6.3Hz, 1H), 2.10 (m, 2H), 1.82–1.55 (m, 2H), 1.53–1.43 (m, 12H); * Equivalent protons with different conformation, and (**b**) FTIR spectra, comparison of the Poly(Ani-AMPS) versus the Ani-AMPS monomer.

**Figure 4 polymers-13-02349-f004:**
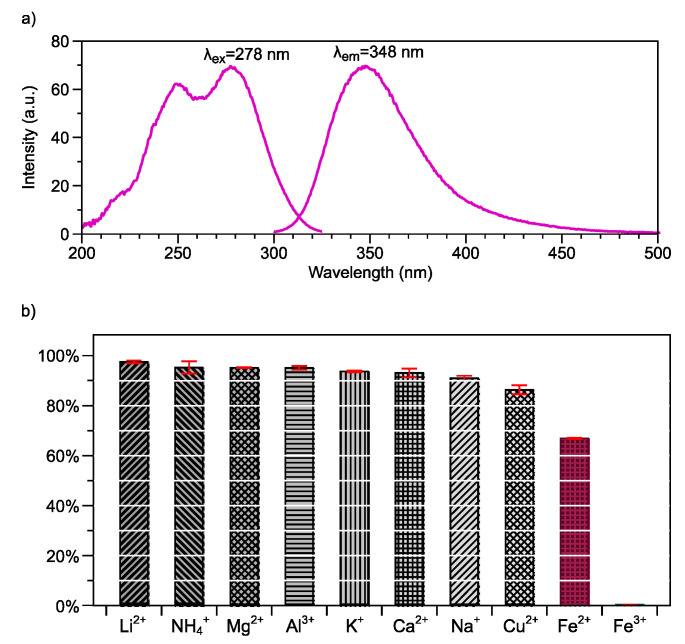
(**a**) Fluorescence spectrum of Poly(Ani-AMPS), (**b**) percentage of emission spectrum turn-off signal evaluating the Poly(Ani-AMPS) interaction with some ions.

**Figure 5 polymers-13-02349-f005:**
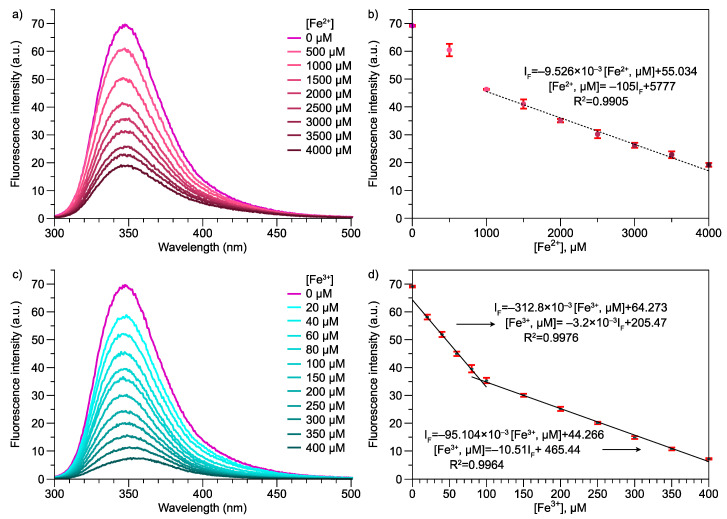
Poly(Ani-AMPS) fluorescence emission spectra regarding the concentration of (**a**) Fe^2+^ and (**c**) Fe^3+^. The plot of the emission intensity at 348 nm vs. the concentration of (**b**) Fe^2+^ and (**d**) Fe^3+^.

**Figure 6 polymers-13-02349-f006:**
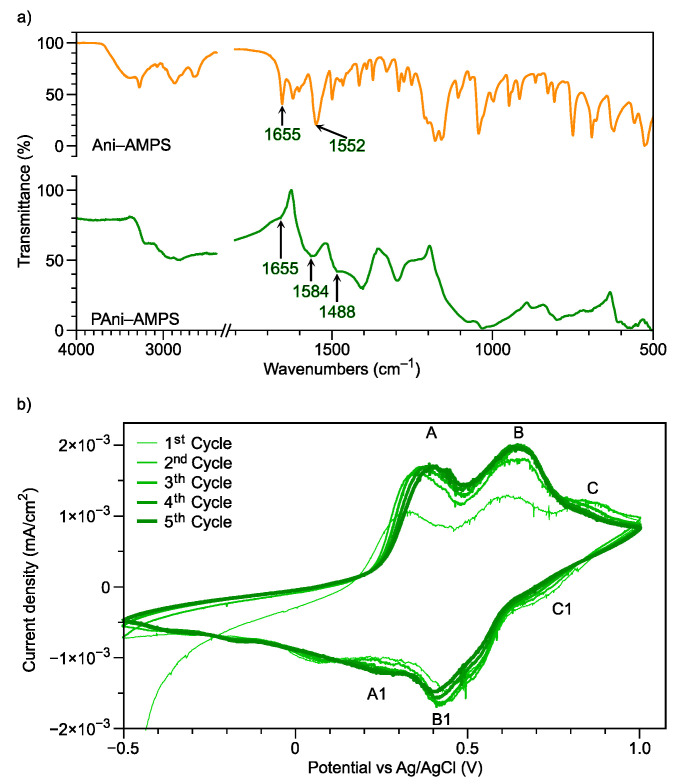
PAni-AMPS characterization by (**a**) FTIR spectroscopy, comparison of PAni-AMPS versus Ani-AMPS spectra, and (**b**) cyclic voltammetry in three-electrode system utilizing sulfuric acid [1 mM] as an electrolyte.

**Figure 7 polymers-13-02349-f007:**
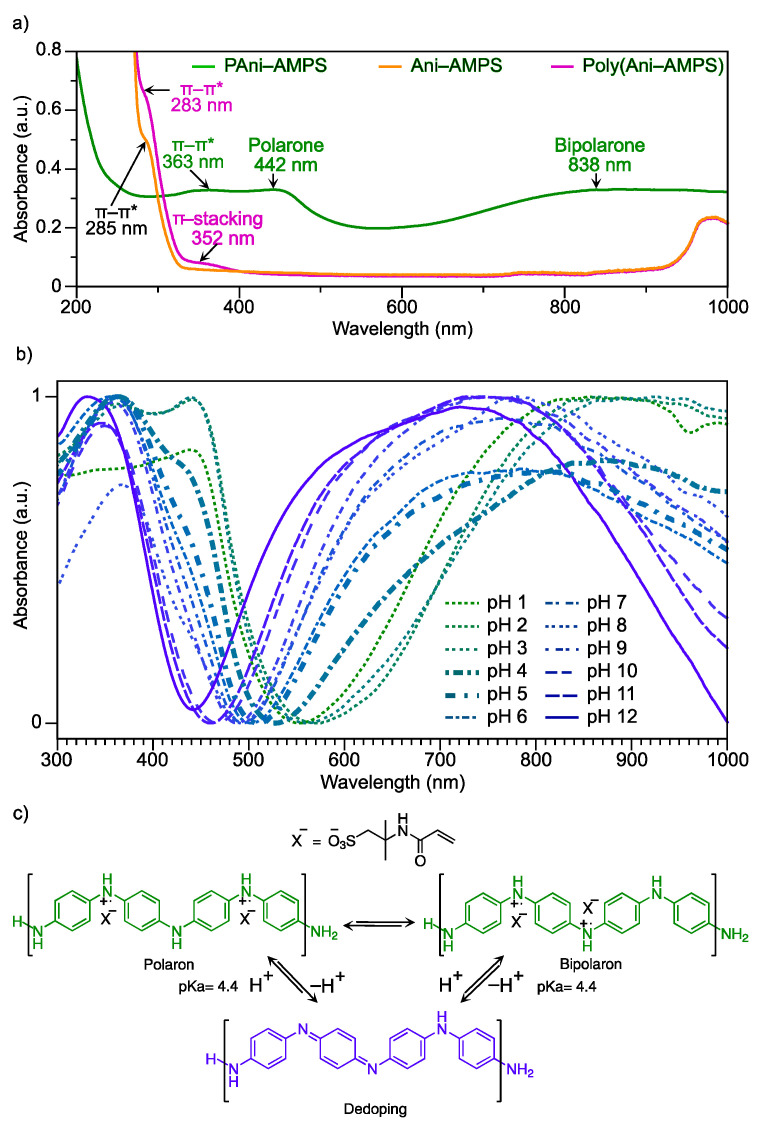
(**a**) UV-Vis-NIR spectra of Ani-AMPS, Poly(Ani-AMPS), and PAni-AMPS; (**b**) UV-Vis-NIR spectra of PAni-AMPS at different pH. (**c**) Scheme of doped-dedoped PAni-AMPS.

**Figure 8 polymers-13-02349-f008:**
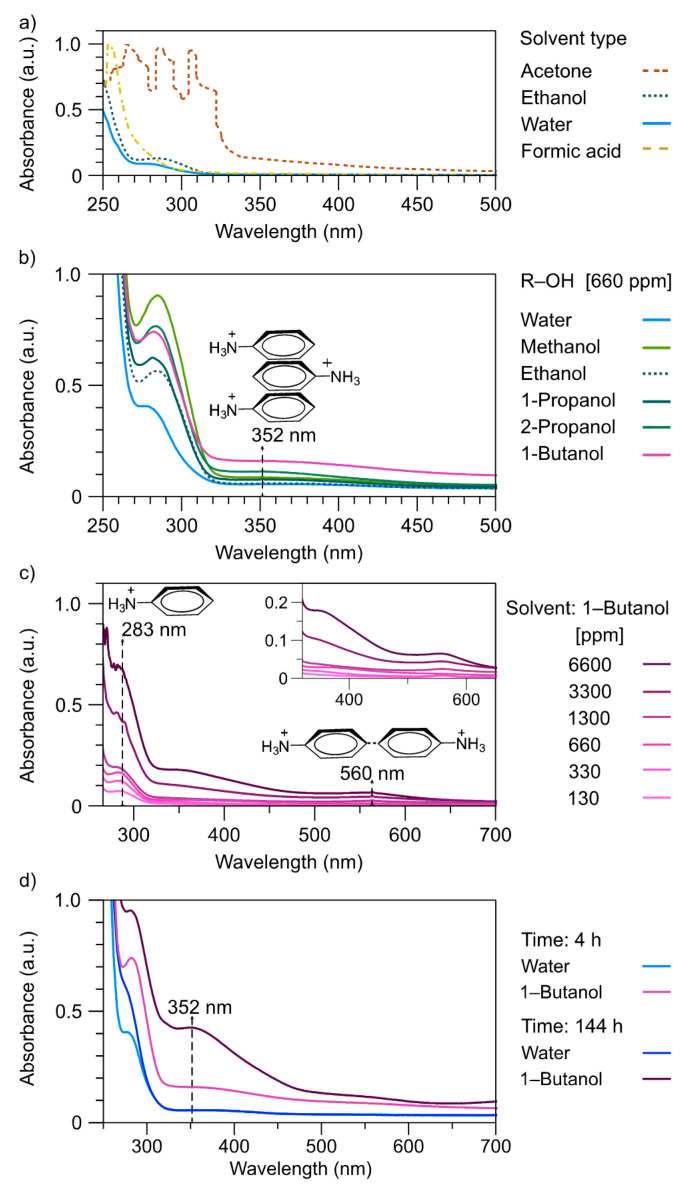
UV-Vis spectra of Poly(Ani-AMPS) in (**a**) different solvents, (**b**) different alcohols, (**c**) different concentrations for 4 h, (**d**) different times (4 and 144 h) at a concentration of 660 ppm.

**Figure 9 polymers-13-02349-f009:**
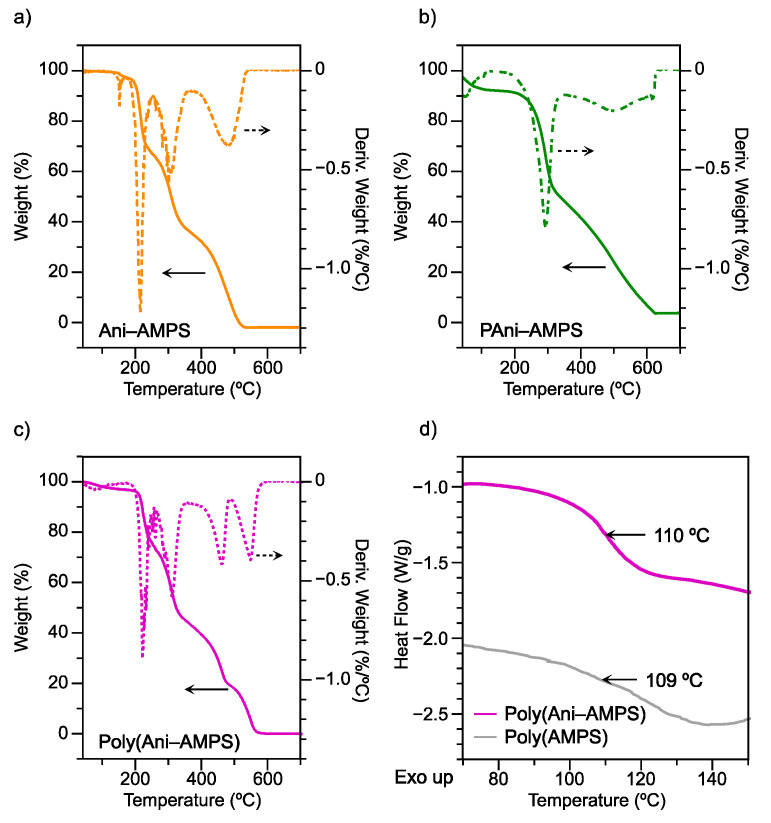
TGA traces and weight derivative of (**a**) Ani-AMPS, (**b**) PAni-AMPS, (**c**) Poly(Ani-AMPS), and (**d**) glass transition temperatures (T_g_) of Poly(Ani-AMPS) and Poly(AMPS-acid).

**Figure 10 polymers-13-02349-f010:**
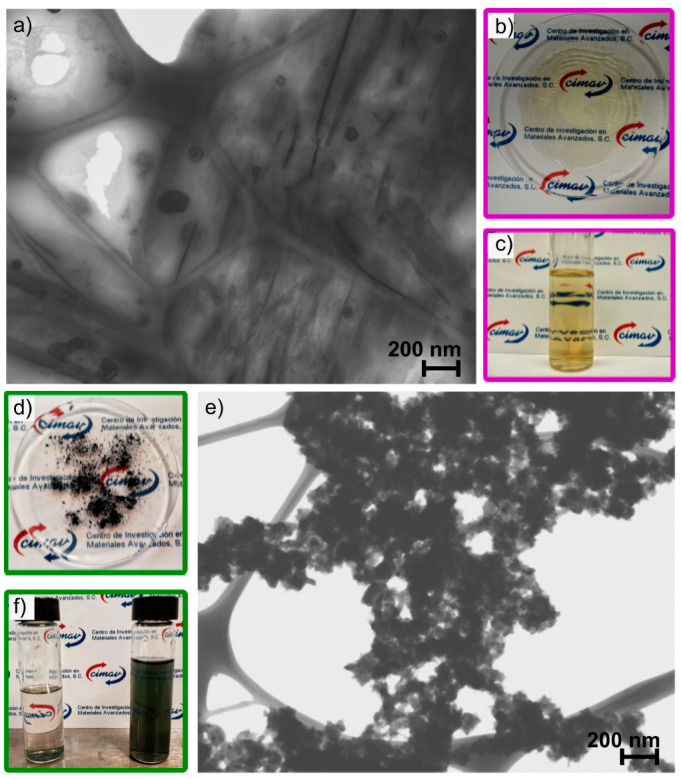
(**a**) Morphology, (**b**) solid phase, and (**c**) water solution of Poly(Ani-AMPS), and (**d**) solid phase, (**e**) morphology, and (**f**) sediment (left) and dispersion (right) in water of PAni-AMPS.

## Data Availability

Not applicable.

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
