# Peer review of "Role of the Anilinium Ion on the Selective Polymerization of Anilinium 2-Acrylamide-2-methyl-1-propanesulfonate"

_polymers, 2021, doi:10.3390/polym13142349_

Round 1
Reviewer 1 Report
The manuscript described an interesting work for selectively polymerizing vinyl group and aniline group to forming polymers with different structures. The structure of polymer was characterized with different methods. The properties of obtained polymer such as fluorescence etc has been investigated. This work is novel enough to be considered for the publication in Polymers. However, some important points must be addressed before that.
- The novelty of this work should be enhanced in introduction part. Also, more references need to be cited in this part. No references have been cited in the first paragraph.
- The molecular weight information of obtained polymers in current manuscript is missed. Molecular weight is one of the key parameter for polymer, it must be provided for the confirming of "polymerization". I wondered for the obtaining of polymer under free radical polymerization due to the existence of proton signals of vinyl protons in Figure 3a. Furthermore the integral value should be provided in the spectra. I am not sure the meaning of "equivalent to two monomers" in line 198-199.
- Some comments need to be provided on "selective polymerization". How to prevent the polymerization of vinyl group in oxidative polymerization using APS as the initiator. How to prevent the polymerization of aniline in free radical polymerization?
Author Response
Response to Reviewer 1 Comments
Point 1: The novelty of this work should be enhanced in introduction part. Also, more references need to be cited in this part. No references have been cited in the first paragraph.
Response 1: Thanks for your comment, the authors changed the introduction.
Point 2: The molecular weight information of obtained polymers in current manuscript is missed. Molecular weight is one of the key parameter for polymer, it must be provided for the confirming of "polymerization". I wondered for the obtaining of polymer under free radical polymerization due to the existence of proton signals of vinyl protons in Figure 3a. Furthermore the integral value should be provided in the spectra. I am not sure the meaning of "equivalent to two monomers" in line 198-199.
Response 2: Thanks for your comments. First, with respect to molecular weight was determined using GPC.
Experimental
2.5 Characterization
The molecular weight of Poly(Ani–AMPS) was determined employing a gel permeation chromatographer (1260 Infinity, Agilent) with a refractive index detector. The equipment has three columns in the molecular weight ranges from 100 to 30,000 g/mol, 10,000 to 200,000 g/mol, and 50,000 to 1,000,000 g/mol. The mobile phase was a pH 8 buffer solution (NaNO3 0.2M, NaH2PO4 0.01M, NaN3 100 ppm). Ten standards samples of poly(ethylene glycol) (Mp = 106 g/mol lower limit, Mp = 21,300 g/mol higher limit) were used for calibration purposes.
- Discussion and results
Lastly, to complement Poly(Ani-AMPS) characterization, the molecular weight was determined by GPC. The Mw obtained was 9,345 g/mol and polydispersity index of 4.39. It is important to note that this molecular weight corresponds to the polyanion due to the pH of the mobile phase.
The second comment, the NMR spectrum was obtained at the end of polymerization for the purpose of determining the conversion, i.e. it is unpurified. However, the polymer was purified with acetone for the purpose of removing these oligomers or monomer without reacting. Concerning the integral of NMR signals, the supplementary material is included for both (monomer, Ani–AMPS and polymer poly(Ani–AMPS)).
Point 3: Some comments need to be provided on "selective polymerization". How to prevent the polymerization of vinyl group in oxidative polymerization using APS as the initiator. How to prevent the polymerization of aniline in free radical polymerization?
Response 3: Thank you for your comment. The authors added an introduction in the results and discussion section.
The selective polymerization that occurs in Ani–AMPS monomer depends mainly on the polymerization conditions. For instance, in free radical polymerization, a redox initiator is utilized, where sodium metabisulfite (MBS) is first placed to produce sodium bisulfite in the presence of water. Thus, salt reacts with ammonium persulfate (APS) to generate free radicals, initiating the polymerization process. It is important to note that sodium bisulfite has a lower oxidation potential than anilinium salts, so APS reacts with sodium bisulfite. In contrast, APS acts as an oxidative agent in oxidative polymerization because the system does not contain MBS. In addition, oxidative polymerization, specifically for PAni–AMPS, requires a molar ratio of 1.25:1 (ammonium persulfate: Ani–AMPS). Compared to free radical polymerization, oxidative polymerization demands an increased amount of initiator as an oxidizing agent. Figure S4 illustrates the scheme of the decomposition of MBS and its reaction with APS.

Reviewer 2 Report
The authors present a manuscript disclosing the role of anilinium ion on a free radical polymerization of an acrylamide derivate. The paper is well done, it has all the necessary part and all the analysis they did are commented and delucidated in detail. I really appreciate the meticulousness with which they suggest all the interpretations of IRs and NMRs.
I think introduction and conlcusions are also in line with the paper and with the whole vision.
I have some minor comments:
-generally when you use an acronym you have to declare it as first use, even if is in an abstract, even in a caption, every part should be self-standing. so please revise line 29,34,
-line 30: the phrase you wrote doesn't have any sense red alone: I'd rephrase explaining why and what you ment for partially polymerized; and quit the ""(A+ or B-) or (A+B-) "" it doen't give more info than the text.
-line 105: six mL , correct to 6 mL
-figure 2: define what are the signals at 1.52, 2.7, 2.8 ppm, even if solvent or impurities, either in the caption, text or SI.
-figures 2,3,... please write every time the solvent you have used for the NMR, the caption has to be self-standing, you should understand without read the text.
Author Response
Response to Reviewer 1 Comments
Point 1. Generally when you use an acronym you have to declare it as first use, even if is in an abstract, even in a caption, every part should be self-standing. so please revise line 29,34,
Response 1. Thank you very much for your comment, the text was corrected.
Point 2. line 30: the phrase you wrote doesn't have any sense red alone: I'd rephrase explaining why and what you ment for partially polymerized; and quit the ""(A+ or B-) or (A+B-) "" it doen't give more info than the text.
Response 2. Thank you for your comment. The authors changed paragraph 1 of the introduction section.
Point 3. line 105: six mL , correct to 6 mL
Response 3. Thank you very much for your comment, the text was corrected, Line 119.
Point 4. figure 2: define what are the signals at 1.52, 2.7, 2.8 ppm, even if solvent or impurities, either in the caption, text or SI.
Response 4. Thank for your observation. The information was collocated in supplementary material.
Figure S1. NMR spectra. a) 1H NMR (400 MHz, Deuterium Oxide) δ 7.64 – 7.54 (m, 3H), 7.45 (dt, J = 8.0, 1.5 Hz, 2H), 6.26 (dd, J = 17.1, 10.1 Hz, 1H), 6.16 (dd, J = 17.1, 1.5 Hz, 1H), 5.72 (dd, J = 10.1, 1.5 Hz, 1H), 3.44 (s, 2H), 1.53 (s, 6H). *By-products of monomer synthesis, specifically oligomers. b) 13C NMR (101 MHz, Deuterium oxide) δ 197.79, 130.99, 130.11, 129.75, 129.20, 126.57, 122.85, 57.15, 51.98, 26.36 ppm.
Point 5. figures 2,3,... please write every time the solvent you have used for the NMR, the caption has to be self-standing, you should understand without read the text.
Response 5. Thank for your observation. The authors modify the figure and caption of both, placing the solvent and the text more explicit.
Figure 2. Ani–AMPS monomer spectra a) 1H NMR (Solvent Deuterium Oxide) δ 7.64 – 7.54 (m, 3H), 7.45 (dt, J = 8.0, 1.5 Hz, 2H), 6.26 (dd, J = 17.1, 10.1 Hz, 1H), 6.16 (dd, J = 17.1, 1.5 Hz, 1H), 5.72 (dd, J = 10.1, 1.5 Hz, 1H), 3.44 (s, 2H), 1.53 (s, 6H) and b) FT-IR spectra, comparison of Ani–AMPS versus aniline and AMPS–acid.
Figure 3. Poly(Ani–AMPS) spectra a) 1H NMR ( Solvent Deuterium Oxide) δ 7.80-7.29 (m, 10H), 3.71 (t, J = 6.3 Hz, 1H), 3.49-3.25 (m, 4H), 2.67 (t, J = 6.3Hz, 1H), 2.10 (m, 2H), 1.82-1.55 (m, 2H), 1.53-1.43 (m, 12H) and b) FTIR spectra, comparison of the Poly(Ani–AMPS) versus the Ani–AMPS monomer.

Round 2
Reviewer 1 Report
The manuscript has been revised sufficiently to be published.